# A Time-Consistency Curriculum for Learning from Instance-Dependent Noisy Labels

## Abstract

Many machine learning algorithms are known to be fragile on simple instance-independent noisy labels. However, noisy labels in real-world data are more devastating since they are produced by more complicated mechanisms in an instance-dependent manner. In this paper, we target this practical challenge of *Instance-Dependent Noisy Labels* by jointly training (1) a model reversely engineering the noise generating mechanism, which produces an *instance-dependent mapping* between the clean label posterior and the observed noisy label; and (2) a robust classifier that produces clean label posteriors. Compared to previous methods, the former model is novel and enables end-to-end learning of the latter directly from noisy labels. An extensive empirical study indicates that the time-consistency of data is critical to the success of training both models and motivates us to develop a curriculum selecting training data based on their dynamics on the two models' outputs over the course of training. We show that the curriculum-selected data provide both clean labels and high-quality input-output pairs for training the two models. Therefore, it leads to promising and robust classification performance even in notably challenging settings of instance-dependent noisy labels where many SoTA methods could easily fail. Extensive experimental comparisons and ablation studies further demonstrate the advantages and significance of the time-consistency curriculum in learning from instance-dependent noisy labels on multiple benchmark datasets.

## 1 Introduction

The training of neural networks can easily fail in the presence of even the simple instance-independent noisy labels since they quickly lead to model overfitting of the noises (Zhang et al., 2017). In practice, however, it is usually challenging to control the labeling quality of large-scale datasets because the labels were generated by complicated mechanisms such as non-expert workers (Han et al., 2020b). An average of 3.3% noisy labels is identified in the test/validation sets of 10 of the most commonly-used datasets in computer vision, natural language, and audio analysis (Northcutt et al., 2021). Moreover, real-world noisy labels are generated in an instance-dependent manner, which is significantly more challenging to address than the most widely studied but oversimplified instance-independent noises, which assume that the noise only depends on the class (Wei et al., 2022).

Two principal methodologies have been developed to address the label noises: (1) detecting samples $(X, \tilde{Y})$ with correct labels $\tilde{Y} = Y$ (empirically, they are the ones with the smallest loss values) and using them to train a clean classifier (Han et al., 2018b; Yu et al., 2019); (2) learning the noise generating mechanism, i.e., a *transition matrix* $T$ defining the mapping between clean label $Y$ and noisy label $\tilde{Y}$ such that $P(\tilde{Y} \mid X) = T^{\top} P(Y \mid X)$, where $P(\cdot \mid X)$ denotes the posterior vector, and then using it to build statistically consistent classifiers (Liu & Tao, 2016; Patrini et al., 2017; Yang et al., 2021). Although both methodologies have achieved promising results in the simplified instance-independent (class-dependent) setting, they have non-trivial drawbacks when applied to the more practical but complicated instance-dependent noises: (1) the "small loss" trick is no longer effective in detecting correct labels (Cheng et al., 2021) because the loss threshold drastically varies across instances and is determined by each transition matrix $T(X)$; (2) the instance-dependent transition matrix $T(X)$ is not identifiable given only the noisy sample and it heavily relies on the estimation of clean label $Y$ in the triple $(X, Y, \tilde{Y})$ (Yang et al., 2021), which is an unsolved challenge in (1).

Therefore, the two learning problems are entangled, i.e., the training of a clean label predictor and the transition matrix estimator depends on each other's accuracy, which substantially relies on the quality of training data $(X, Y, \tilde{Y})$. Specifically, the "small loss" trick cannot provide a high-quality estimation of $Y$ due to the instance-specific threshold of loss. Moreover, the estimation of $Y$ can change rapidly due to the non-stationary loss, which can fluctuate during training and provide inconsistent training signals over time for both models if selected for training. Furthermore, the data subset selection inevitably introduces biases toward easy-to-fit samples and degrades the data diversity (Yang et al., 2021; Cheng et al., 2021; Berthon et al., 2021; Cheng et al., 2020), which in fact is critical to the training and the accuracy of both models, especially the transition matrix estimator, because easy-to-fit samples usually have extremely sparse transition matrices.

To tackle the above issues, we propose a novel metric "Time-Consistency of Prediction (TCP)" to select high-quality data to train both models. TCP measures the consistency of model prediction for an instance over the course of training, which reflects whether its given label results in gradients consistent with the majority of other instances, and this criterion turns out to be a more reliable identifier of clean labels. When applied to the training of clean label predictor, TCP is more accurate in clean label detection than "small loss" (or high confidence) criterion, because it avoids the comparison of confidence for samples with instance-dependent loss/confidence thresholds. Moreover, when applied to the training of the transition matrix estimator, TCP measures the time-consistency of predicted noisy labels $\tilde{Y}$. Surprisingly, it also faithfully reflects the correctness of the predicted clean label $Y$. Since the objective to estimate the transition matrix is defined by both $Y$ and $\tilde{Y}$, selecting samples with high TCP considerably improves the training of the transition matrix estimator. In addition, to exploit the data diversity in training the two models, we apply a curriculum that starts from selecting only a few high TCP data for early-stage training but progressively includes more training data once the two models become maturer and more consistent.

In this paper, we develop a three-stage training strategy with the TCP curriculum embedded. In every training step, we first update the clean label predictor using selected data with high TCP on this model, followed by training the transition matrix estimator given the predicted clean label posterior and the noisy labels on selected data with high TCP on the estimator, and end with fine-tuning the clean label predictor directly using the noisy label and the estimated transition matrix. It is worth noting that the TCP metrics for the two models are updated using the model outputs collected from this dynamic training process without causing additional cost. As demonstrated by extensive empirical studies and experimental comparisons, our method leads to efficient joint training of the two models that mutually benefits from each other and produces an accurate estimation of both the clean label and instance-dependent transition matrix. On multiple benchmark datasets with either synthetic or real-world noises, our method achieves state-of-the-art performance with significant improvements.

## 2 BACKGROUNDS AND RELATED WORKS

Let $(X, Y) \in \mathcal{X} \times \{1, \ldots, c\}$ be the random variables for instances and clean labels, where $\mathcal{X}$ represents the instance space and $c$ is the number of classes. In many real-world applications, the observed labels are not always correct but contain some noise. Let $\tilde{Y}$ be the random variable for the noisy label. What we have is a sample $\{(x_1, \tilde{y}_1), \ldots, (x_n, \tilde{y}_n)\}$ drawn from the noisy distribution $\mathcal{D}_\rho$ of the random variables $(X, \tilde{Y})$. We aim to learn a robust classifier that could assign clean labels to test data by exploiting the sample with noisy labels.

**Label noise models.** Currently, there are three typical label noise models, which are the random classification noise (RCN) model (Biggio et al., 2011; Natarajan et al., 2013; Manwani & Sastry, 2013), the class-dependent label noise (CDN) model (Patrini et al., 2017; Xia et al., 2019; Zhang & Sabuncu, 2018), and the instance-dependent label noise (IDN) model (Berthon et al., 2021; Cheng et al., 2021). Specifically, RCN assumes that clean labels flip randomly with a constant rate; CDN assumes that the flip rate only depends on the true class; IDN considers the most general case of label noise, where the flip rate depends on its instance. Since IDN is non-identifiable without any additional assumption, some simplified variants were proposed. (Xia et al., 2020b) proposed the part-dependent label noise (PDN) model which assumes that the label noise depends on parts of instances. (Cheng et al., 2020; Yang et al., 2021) assume that the flip rates are dependent on instances but can be upper bounded by a value smaller than 1. This paper focuses on the original IDN model without introducing any additional assumptions.

**Estimating the transition matrix.** The structure of label noise is usually formulated by a $c \times c$ transition matrix $T$, where $c$ is the number of classes and its $ij$-th element $T_{ij}(x) = P(\tilde{Y} = j \mid Y = i, X = x)$, which represents the probability that the instance $x$ with the clean label $Y = i$ actually has a noisy label $\tilde{Y} = j$. The transition matrix naturally establishes the connection between noisy posterior and clean posterior, i.e., $P(\tilde{Y} \mid X) = T^{\top}(X)P(Y \mid X)$, and thus plays an important role in building statistically consistent classifiers in label-noise learning (Liu & Tao, 2016; Scott, 2015). To estimate it, a cross-validation method can be applied for the binary classification task (Natarajan et al., 2013). For CDN, the transition matrix could be learned by exploiting anchor points (Patrini et al., 2017; Yu et al., 2018). For IDN, the transition matrix for an instance could be approximated by a combination of the transition matrices for the parts of the instance (Xia et al., 2020b) or a Bayes label transition matrix (Yang et al., 2021). Yao et al. (2021) exploited the causal graph to estimate the transition relations between clean and noisy labels.

**Curriculum learning.** Curriculum learning was first proposed by Bengio et al. (2009), which describes a learning paradigm in which a model is learned by gradually introducing samples of increasing hardness to training. Its effectiveness has been empirically verified in a wide range of applications, e.g., computer vision (Chen & Gupta, 2015), natural language processing (Turian et al., 2010), and multitask learning (Graves et al., 2017). Curriculum for label-noise learning has been also investigated. MentorNet (Jiang et al., 2018) pre-trains an extra network producing a data-driven curriculum selecting data instances to guide the training. When the clean validation data is not available, MentorNet has to use a predefined curriculum. RoCL (Zhou et al., 2021) develops a curriculum learning strategy that smoothly transitions between (1) detection and supervised training on clean data; and (2) relabeling and self-supervision on noisy data. Nevertheless, RoCL has no convergence guarantee and needs extra data augmentations to collect spatial-consistent pseudo labels.

Moreover, existing methods for learning with noisy labels employ heuristics to reduce the side-effect of noisy labels, e.g., selecting reliable samples (Han et al., 2018b; Yu et al., 2019; Wei et al., 2020a; Wu et al., 2020; Xia et al., 2020a), reweighting samples (Ren et al., 2018; Jiang et al., 2018; Ma et al., 2018; Kremer et al., 2018; Reed et al., 2015), correcting labels Tanaka et al. (2018); Zheng et al. (2020), designing robust loss functions Zhang & Sabuncu (2018); Xu et al. (2019); Liu & Guo (2020); Ma et al. (2020), employing side information (Vahdat, 2017; Li et al., 2017), and (implicitly) adding regularization (Li et al., 2021; 2017; Veit et al., 2017; Vahdat, 2017; Han et al., 2018a; Zhang et al., 2018; Guo et al., 2018; Hu et al., 2020; Zhang et al., 2021; Han et al., 2020a).

## 3 EXAMPLES SELECTION CRITERION: TIME-CONSISTENCY OF PREDICTION

According to the observation that the loss on instances with clean labels is usually smaller than instances with noisy labels, the loss computed at an instantaneous step has been widely adopted as a selection criterion for confident examples (Han et al., 2018b; Yu et al., 2019; Wang et al., 2019). It is because instances with clean labels are mutually consistent with each other in producing gradient updates, allowing the model to fit them better and thereby make the loss smaller than instances with noisy labels.

Unfortunately, the instantaneous loss was found only work well on the instance-independent label noise (Cheng et al., 2021). For a deep neural network, because of the non-smooth nature of the loss and the randomness of stochastic gradient descent, the instantaneous loss of each instance can change dramatically between consecutive epochs, leading to a huge gap between training sets selected over consecutive epochs. Therefore, it is necessary to take the training history of each instance into consideration. Zhou et al. (2020) proposed a robust version of the instantaneous loss as the exponential moving average of it over the course of training. Nevertheless, in the IDN case, each instance with its noisy label is a unique pattern, which is more complex and thereby requires a more robust selection criterion. Apparently, at the instance level, the one-hot prediction of an instance is a more robust metric than the loss because the former has a tolerance to the change of predicted class posterior while the latter has not, i.e., the one-hot prediction remains unchanged if the position of the max element in the predicted class posterior vector is maintained but the cross-entropy loss changes once the predicted class posterior changes.

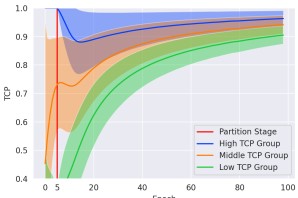 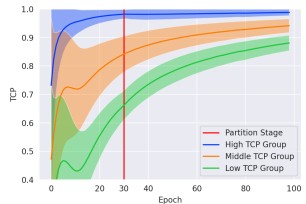 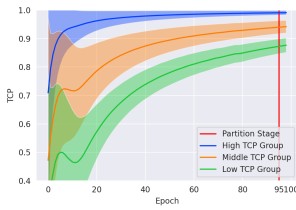

(a) Partitioned at a start stage.    (b) Partitioned at a early stage.    (c) Partitioned at an end stage.

Figure 1: TCP (mean and std.) of three groups (high TCP (10%), middle TCP (80%), and low TCP (10%)) partitioned by the TCP calculated at the start stage (epoch 5), early stage (epoch 30), and end stage (epoch 95) during training a ResNet34 on CIFAR10 with IDN-0.4 for 100 epochs.

Inspired by the above insights, we propose a time-consistency of prediction (TCP) metric as follows:

$$\text{TCP}_{t+1}(x) = \frac{t}{t+1}\text{TCP}_t(x) + \frac{1}{t+1}\text{InP}_{t+1}(x), \tag{1}$$

where $\text{InP}_{t+1}(x) = \mathbb{1}[\hat{y}_{t+1} = \hat{y}_t]$ and $\hat{y}_t$ is the predicted label at epoch $t$. This metric considers the prediction consistency over the course of training, which can better describe the IDN data and select confident examples than the previous ones. To see this, we first manually add IDN at 0.4 noise rate (see Section. 5) onto a benchmark dataset *CIFAR10* and train a ResNet34 (He et al., 2016) for 100 epochs with a constant learning rate. Since no curriculum strategy is applied here, we select confident examples at every epoch $t$ with a fixed number 5,000 according to four types of selection criterion, i.e., instantaneous prediction $\text{InP}_t(x)$, instantaneous loss $\ell(x)$, time-consistency of prediction $\text{TCP}_t(x)$, and time-consistency of loss (defined in the same way as TCP). Then we count the number of instances with clean labels from the selected confident examples and calculate the clean ratios (ratio of clean instances to selected instances). As shown in Figure 2, we can find that the two instantaneous metrics have clean ratios lower than 0.6, which are worse than random selection. As for time-consistency of loss, the clean ratio is slightly higher than the random selection. Those three metrics are basically not discriminative to the noisy data. By contrast, the proposed TCP metric has a distinguishable performance, uplifting more than 20 percent of the clean ratio of the selected confident examples. More empirical study regarding Figures 1 and 2 is provided in Appendix A.

Moreover, we partition the whole data into three groups (high TCP (10%), middle TCP (80%), and low TCP (10%)) by the TCP calculated at the start stage (epoch 5), early stage (epoch 30), and end stage (epoch 95). We visualize the mean and variance of the groups through the whole training epochs. As shown in Figure 1, the start-stage partition fails at the end stage as three groups are entangled together while the early-stage partition shares the almost same pattern as the end-stage partition. Thus, we can conclude that the early-stage TCP is reflective of the property of each instance in the future, which means the time-consistent examples selected in the early stage will not mislead the classifier because their TCP are still high and thus they will still be selected as time-consistent examples in the rest training epochs. Besides, a warmup for the TCP is proved to be necessary.

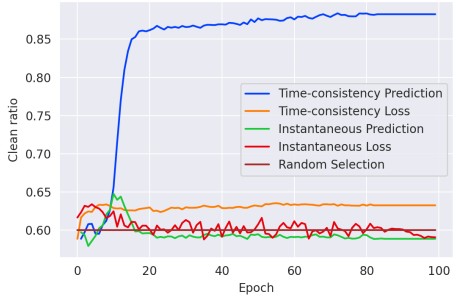

Figure 2: Clean ratios of the selected top 5000 instances ranked by four kinds of instance hardness measures, respectively, during a standard training for 100 epochs. The clean ratio of randomly selected instances is 0.6 since the noise rate is 0.4.

## 4 TCP GUIDED CURRICULUM LEARNING FOR INSTANCE-DEPENDENT NOISY LABELS

Our curriculum contains two sub-curriculum corresponding to two main challenges for solving the IDN problem: (1) detecting examples with clean labels; (2) learning the transition matrix. In this

section, we elaborate on how to use TCP to design curricula to achieve these goals. Below we use clean- and noisy-TCP to refer to time consistency of clean label predictions ($\arg\max \hat{P}(y \mid x)$) and noisy label predictions ($\arg\max \hat{P}(\tilde{y} \mid x)$) over historical training steps, respectively.

### 4.1 LEARNING A PRIMARY CLEAN CLASSIFIER WITH HIGH CLEAN-TCP INSTANCES

In Section 3, we demonstrate that clean-TCP can be used to select examples with clean labels because they are mutually consistent with each other in producing gradient updates and easy to be learned to have high clean-TCP. However, as shown in Figure 2, not all of the selected high clean-TCP instances have clean labels. The reason why those instances have high clean-TCP but not clean labels is because they are far away from the classification boundary and inherently hard to be misclassified even though their given labels are wrong. Namely, their labels get corrected by the classifier, and high clean-TCP indicates the pseudo labels assigned by the classifier are correct.

Therefore, with clean-TCP, we design a curriculum which exploits both examples with clean labels and instances with correct pseudo labels to learn a clean classifier. We theoretically prove that introducing high clean-TCP instance with its pseudo label does not cause catastrophic forgetting of the learned confident examples. Consider the situation we have a labeled set $\mathcal{L}$ (in practice it can be the selected confident examples set) and one unlabeled instance $x'$. By training on $\mathcal{L}$ for one step, we have $\theta_{t+1} = \theta_t + \eta \sum_{x \in \mathcal{L}} \nabla_\theta \ell(x; \theta_t)$; and by training on $\mathcal{L}$ and $x'$ for one step, we have $\theta'_{t+1} = \theta_t + \eta \left( \sum_{x \in \mathcal{L}} \nabla_\theta \ell(x; \theta_t) + \nabla_\theta \ell(x'; \theta_t) \right)$, where $\theta_t$ denotes the network parameters at step $t$ and $\eta$ denotes the learning rate. Then we have

$$\frac{1}{\eta} \left| \sum_{x \in \mathcal{L}} \left[ \ell(x; \theta_{t+1}) - \ell\left(x; \theta'_{t+1}\right) \right] \right| = \left| \frac{p_{t+1}^{\hat{y}'_t}(x')}{p_t^{\hat{y}'_t}(x')} - 1 \right|, \tag{2}$$

where $p_t^{\hat{y}'_t}(x')$ is the probability of $x'$ belonging to $\hat{y}'_t$ at step $t$, and $\hat{y}'_t$ is the prediction (pseudo label) of $x'$ at step $t$. The detailed derivation is provided in Appendix B. If $x'$ is selected with high clean-TCP, $p_t^{\hat{y}'_t}(x')$ is very close to $p_{t+1}^{\hat{y}'_t}(x')$ because it has been verified in Figure 1 that instances with high clean-TCP in the early stage maintain their high clean-TCP in the future, which means the loss change can be bounded with a very small value. Therefore, exploiting high clean-TCP instances with pseudo labels helps to correct corrupted labels and learn a clean classifier without causing catastrophic forgetting of the learned examples with correct labels. In Figure 3, we show the clean ratios of the original noisy labels and pseudo labels of instances selected with our curriculum during the whole training process. The clean ratio for

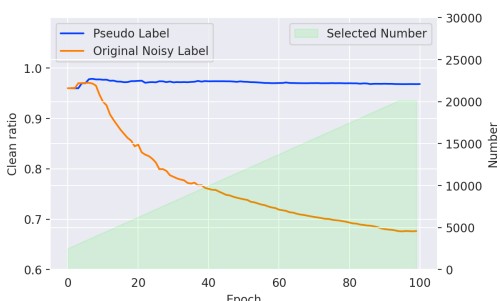

Figure 3: Clean ratios of selected high clean-TCP examples w.r.t. their original noisy labels and pseudo labels with linear growth of the selected number during our curriculum learning on CIFAR10 with IDN-0.4 for 100 epochs.

pseudo labels maintains an amazing high value, much better than the original clean ratio. Therefore, the clean classifier can be learned by minimizing $\sum_{n=1}^{N} \mathcal{L}\left(f(x_n), y_n^*\right)$, where $y^*$ can be the original noisy label or pseudo label in different learning phases. Implementation details can be found in Section 4.3.

### 4.2 LEARNING A TRANSITION MATRIX WITH HIGH NOISY-TCP INSTANCES

The transition matrix is not identifiable by only exploiting noisy data without introducing additional assumptions, therefore we formulate the objective function for leaning the transition matrix based on the equation $P(\tilde{Y} \mid X) = T^\top(X) P(Y \mid X)$ as follow:

$$\min_T \quad \frac{1}{N} \sum_{n=1}^{N} \mathcal{L}\left(T^\top(x_n) f(x_n), \tilde{y}_n\right), \qquad \text{where } f(x_n) = \hat{P}(y \mid x_n). \tag{3}$$

To select the high-quality triplets $(X, Y, \tilde{Y})$ for the above objective, two conditions should be considered. First, it is necessary for $f(\cdot)$ to output a precise clean class posterior, otherwise, $T$ cannot be optimized in the correct direction, in the case $\tilde{y}$ is given and $f(\cdot)$ has been learned in advance and is fixed. As we discussed above, instances with high clean-TCP tend to have the correct pseudo label, and thereby a precise clean class posterior, which satisfies this necessary condition. Second, the noisy-TCP should be high. By treating $T^{\top}(x)f(x)$ as a whole predictor for $\tilde{y}$, the corresponding new objective is to predict $\tilde{y}$. Therefore, high noisy-TCP instances naturally indicate the instance is learned better and faster for predicting $\tilde{y}$, leading to stable and fast learning.

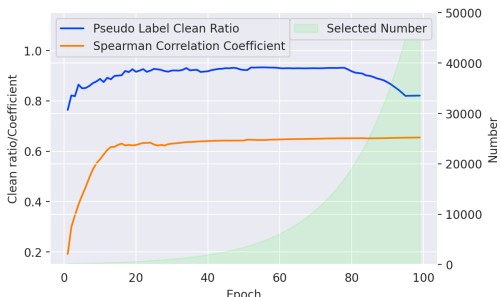

Figure 4: Clean ratios of selected high noisy-TCP examples w.r.t. their pseudo labels with exponential growth of the selected number and Spearman rank-order correlation coefficient between the noisy- and clean-TCP during our curriculum learning on CIFAR10 with IDN-0.4 for 100 epochs.

We discover that the noisy-TCP inherently has a strong correlation with clean-TCP so we can use it to select triplets fulfilling both conditions above. To see this, at each epoch, we calculate the Spearman rank-order correlation coefficient[1] between the noisy-TCP and clean-TCP of the whole dataset. Besides, we calculate the clean ratio of the selected high noisy-TCP instances w.r.t. their pseudo labels. In Figure 5, we show the data distribution in terms of clean- and noisy-TCP at epoch 50 (More data distributions at different epochs are provided in Appendix C). The green regression line partially implies the linear correlation between the clean- and noisy-TCP. Also, instances with correct pseudo labels are mainly distributed in the high TCP area, and vice versa. As shown in 4, the Spearman rank-order correlation coefficient is above 0.6 after 10 epochs with a consistent 0 $p$-value, roughly indicating that noisy-TCP is strongly Spearman rank-order correlated with clean-TCP for 100% sure. Meanwhile, the clean ratio is consistently above 0.8, which means those high noisy-TCP instances also have correct clean predictions and

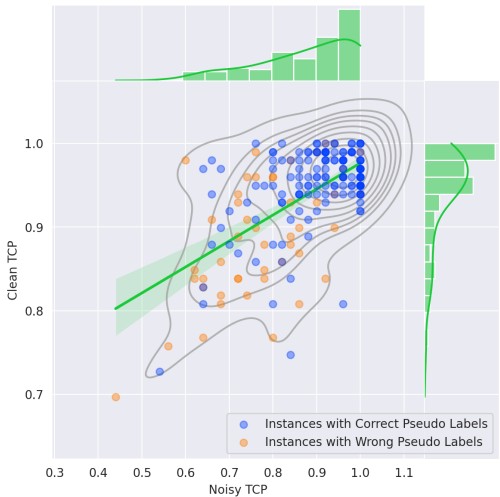

Figure 5: Data distribution in terms of noisy- and clean-TCP at epoch 50 during our curriculum learning on CIFAR10 with IDN-0.4 for 100 epochs.

thereby probably precise clean class posterior. Note that the clean ratio decreases at the late stage because the curriculum selects almost all the data at the end. Overall, high noisy-TCP instances not only are naturally stable for the new objective to predict $\tilde{y}$ but also satisfy the necessary condition to have precise clean class posterior, which make them perfect examples for learning the transition matrix.

### 4.3 TCP GUIDED CURRICULUM LEARNING ALGORITHM

The main steps of our algorithm are summarized in Algorithm 1 with the complete procedure detailed in Appendix D. First, we warm up the feature extractor $\phi$, classification layer $c$ by minimizing a standard cross-entropy (CE) loss on noisy data, and meanwhile compute the clean-TCP for every instance. Then, we warm up the transition matrix layer $t$ with high clean-TCP instances and obtain the noisy-TCP for every instance. From now on, iteratively, high clean-TCP instances are fed to the clean classifier (green part in Figure 6) to train a primary clean classifier with the clean CE loss, and based on the primary clean classifier, instances with high noisy-TCP are fed to the transition matrix (blue part in Figure 6) to train a transition matrix with the noisy CE loss while the parameters of

---

[1]The Spearman rank-order correlation coefficient is a nonparametric measure of the monotonicity of the relationship between two sets (Zar, 2005).

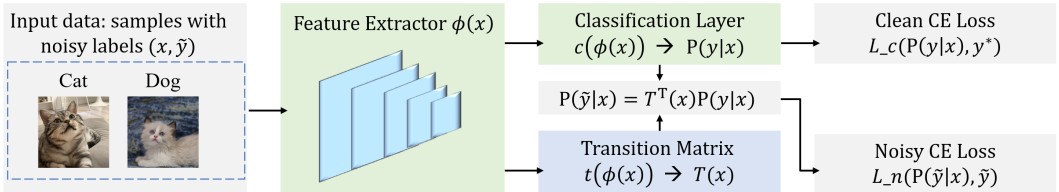

Figure 6: An overview of the proposed method. The second image of cat has a noisy label as "dog". The transition matrix $T(\cdot) = t(\phi(\cdot))$ and classifier $f(\cdot) = c(\phi(\cdot))$ share a common feature extractor.

---

**Algorithm 1** TCP Guided Curriculum Learning Algorithm.

---

**Input:** Noisy training sample $\mathcal{D}$, feature extractor $\phi$, classification layer $c$, transition matrix layer $t$, $f(\cdot) = c(\phi(\cdot))$, $T(\cdot) = t(\phi(\cdot))$, number sequences $N_c$ and $N_t$, training epoch $e$.

1: **Warmup clean- and noisy- TCP**
2: **Curriculum training:**
3: **for** $e$ in $\{1, \cdots, e\}$ **do**
4:      Select $N_t[e]$ high noisy-TCP instances as $\mathcal{D}_t$.
5:      Train $t$ while fixing $\phi$ and $c$ on $\mathcal{D}_t$ by minimizing $\sum_{n=1}^{N} \mathcal{L}_n \left( T^{\top}(x_n) f(x_n), \tilde{y}_n \right)$.
6:      Select $N_c[e]$ high clean-TCP instances as $\mathcal{D}_c$.
7:      Train $\phi$ and $c$ on $\mathcal{D}_c$ by minimizing $\sum_{n=1}^{N} \mathcal{L}_c \left( f(x_n), y_n^* \right)$, where $y^*$ is the pseudo label.
8:      Fix $t$ and fine-tune $\phi$ and $c$ on $\mathcal{D}$ by minimizing $\sum_{n=1}^{N} \mathcal{L}_n \left( T^{\top}(x_n) f(x_n), \tilde{y}_n \right)$.
9:      Record the clean and noisy prediction and calculate the clean- and noisy-TCP by Eq. equation 1.
10: **end for**

**Output:** Optimized feature extractor $\phi$, classification layer $c$, transition matrix layer $t$.

---

the primary clean classifier are frozen. Then the clean classifier gets improved by being fine-tuned on the whole data with the fixed transition matrix. The clean- and noisy-TCP of every instance are updated at the end of each epoch. Finally, a transition matrix with a small estimation error and a clean classifier with a performance improvement can be obtained.

## 5 EXPERIMENTS

In this section, we examine how the proposed methods learn a robust classifier against instance-dependent noisy labels.

**Dataset.** We employ three widely used datasets, i.e., *F-MNIST* (Xiao et al., 2017), *SVHN* (Netzer et al., 2011), and *CIFAR10/100* (Krizhevsky et al., 2009), and four versions of the real-world noisy dataset *CIFAR10N* (Wei et al., 2022), and *Clothing1M* (Xiao et al., 2015). F-MNIST contains 60,000 training images and 10,000 test images with 10 classes. SVHN and CIFAR10 both have 10 classes of images, but the former contains 73,257 training images and 26,032 test images, and the latter contains 50,000 training images and 10,000 test images while CIFAR100 has 100 classes. CIFAR10N provides CIFAR10 images with human-annotated noisy labels obtained from Amazon Mechanical Turk. Four versions of CIFAR10N label sets are employed here, three of which are labeled by three independent workers (named *CIFAR10N-1/2/3*) and one of which is negatively aggregated from the above three sets (named *CIFAR10N-W*). Clothing1M has 1M images with real-world noisy labels and additional 50k, 14k, 10k images with clean labels for training, validation and test, and we only use noisy training set in the training phase. For all the datasets, we leave out 10% of the training data as a validation set, which is for model selection. The final test model is selected with the highest validation accuracy.

**Noisy labels generation.** For clean datasets, we artificially corrupt the class labels of training and validation sets following the instance-dependent noisy labels generalization method in Xia et al. (2020b). We generate noisy datasets of {0.1, 0.2, 0.3, 0.4, 0.5} five noise rates.

Table 1: Means and stds of classification accuracy on *CIFAR10* with different label noise rates.

| | IDN-0.1 | IDN-0.2 | IDN-0.3 | IDN-0.4 | IDN-0.5 |
|---|---|---|---|---|---|
| CE | 74.49±0.29 | 68.21±0.72 | 60.48±0.62 | 49.84±1.27 | 38.86±2.71 |
| Decoupling | 74.09±0.78 | 70.01±0.66 | 63.05±0.65 | 44.27±1.91 | 38.63±2.32 |
| MentorNet | 74.45±0.66 | 70.56±0.34 | 65.42±0.79 | 46.22±0.98 | 39.89±2.62 |
| Co-teaching | 76.99±0.17 | 72.99±0.45 | 67.22±0.64 | 49.25±1.77 | 42.77±3.41 |
| Co-teaching+ | 74.27±1.20 | 71.07±0.77 | 64.77±0.58 | 47.73±2.32 | 39.47±2.14 |
| Joint | 76.89±0.37 | 73.89±0.34 | 69.03±0.79 | 54.75±5.98 | 44.72±7.72 |
| DMI | 75.02±0.45 | 69.89±0.33 | 61.88±0.64 | 51.23±1.18 | 41.45±1.97 |
| Forward | 73.45±0.23 | 68.99±0.62 | 60.21±0.75 | 47.17±2.96 | 40.75±2.09 |
| Reweight | 74.55±0.23 | 68.42±0.75 | 62.58±0.46 | 50.12±0.96 | 41.08±2.45 |
| T-Revision | 74.61±0.39 | 69.32±0.64 | 64.09±0.37 | 50.38±0.87 | 42.57±3.27 |
| CRUST | 76.20±0.90 | 76.41±1.56 | 71.06±2.21 | 64.59±3.32 | 52.50±0.81 |
| CausalINL | 79.08±0.40 | 75.65±1.04 | 67.70±0.82 | 49.19±0.82 | 47.83±1.58 |
| PTD | 78.71±0.22 | 75.02±0.73 | 71.86±0.42 | 56.15±0.45 | 49.07±2.56 |
| Bayes | 80.17±1.32 | 79.51±1.21 | 76.43±1.99 | 69.53±3.24 | 57.42±4.37 |
| TCP | **82.49±0.51** | **80.88±0.90** | **79.32±1.59** | **76.03±2.58** | **60.66±5.71** |

**Baselines and measurements.** On synthetic noisy datasets, without introducing data augmentation techniques and semi-supervised learning, we compare the proposed method TCP with the following baselines: (i). CE, which optimizes the standard cross-entropy loss on noisy datasets. (ii). Decoupling (Malach & Shalev-Shwartz, 2017), which trains two networks on samples whose predictions are different. (iii). MentorNet Jiang et al. (2018), Co-teaching (Han et al., 2018b), and Co-teaching+ (Yu et al., 2019), that mainly handle noisy labels by training on instances with small loss. (iv). Joint (Tanaka et al., 2018), which jointly optimizes labels and network parameters. (v). DMI (Xu et al., 2019), which uses a novel information-theoretic loss function to learn a robust classifier. (vi). Forward (Patrini et al., 2017), Reweight (Liu & Tao, 2016), and T-Revision (Xia et al., 2019), that utilize a class-dependent transition matrix $T$ to correct the loss function. (vii). PTD (Xia et al., 2020b) and Bayes (Yang et al., 2021), estimate instance-dependent transition matrix under some additional assumptions; CRUST (Mirzasoleiman et al., 2020) iteratively selects subsets of clean data points that provide an approximately low-rank Jacobian matrix; CausalINL (Yao et al., 2021) exploits the causal graph to estimate the transition relations between clean and noisy labels. On real-world noisy datasets, we apply the transition matrix learning and fine-tuning parts to the SoTA method Dividemix (Li et al., 2020), i.e., at each epoch, in addition to the Dividemix training, we select high noisy-TCP data to learn the transition matrix and use it to fine-tune the whole data. Then we compare this combined method **TCP-D** with the following SoTA methods: (i). PES (Bai et al., 2021). (ii). Dividemix (Li et al., 2020). (iii). CORES (Cheng et al., 2021). (iv). ELR+ (Liu et al., 2020). (v). JoCoR (Wei et al., 2020b). (vi). CAL (Zhu et al., 2021). We use a ResNet18 network for F-MNIST, a ResNet34 network for SVHN and CIFAR10, a ResNet50 network for CIFAR100, a PreAct-ResNet18 for CIFAR10N, and a pre-trained ResNet50 network for Clothing1M. More training details are provided in Appendix E. Classification accuracy is employed to evaluate the performance of each model on the clean test set. Results over 5 trials on all datasets except Clothing1M, for which the result is over 1 trial, are reported.

**Comparison with the State-of-the-Arts.** We compare TCP with multiple baselines using the same network architecture. Table 1 show the results on CIFAR10 with different rates of IDN from 0.1 to 0.5, respectively. TCP outperforms baselines across all datasets and noise rates. The improvement is significant when the noise rate is large. Tables 5, 6, and 7 show the results on F-MNIST, SVHN, and CIFAR100, which are provided in Appendix F. Table 2 show the results on real-world noisy datasets CIFAR10N-1/2/3/W and Clothing1M. TCP-D consistently achieves the best test accuracy on real-world noisy datasets. Note that the results of baselines on CIFAR10N are taken from the official leaderboard http://www.noisylabels.com/.

**Comparison on the transition matrix estimation error.** We compare the transition matrix estimation error of our method with the instance-independent method Forward (Patrini et al., 2017), and two

Table 2: Means and stds of classification accuracy on real-world noisy datasets.

| | CIFAR10N-1 | CIFAR10N-2 | CIFAR10N-3 | CIFAR10N-W | Clothing1M |
|---|---|---|---|---|---|
| PES (semi) | 95.06±0.15 | 95.19±0.23 | 95.22±0.13 | 92.68±0.22 | 74.29 |
| DivideMix | 95.16±0.19 | 95.23±0.07 | 95.21±0.14 | 92.56±0.42 | 74.30 |
| CORES | 94.45±0.14 | 94.88±0.31 | 94.74±0.03 | 91.66±0.09 | 73.24 |
| ELR+ | 94.43±0.41 | 94.20±0.24 | 94.34±0.22 | 91.09±1.60 | 74.31 |
| JoCoR | 90.30±0.20 | 90.21±0.19 | 90.11±0.21 | 83.37±0.30 | 70.30 |
| CAL | 90.93±0.31 | 90.75±0.30 | 90.74±0.24 | 85.36±0.16 | 74.21 |
| TCP-D | **95.51±0.06** | **95.37±0.08** | **95.43±0.04** | **93.36±0.09** | **74.41** |

| | IDN-0.2 | IDN-0.4 |
|---|---|---|
| TCP | **80.88±0.90** | **76.03±2.58** |
| TCP w/o $\mathcal{D}_c$ | 79.94±0.86 | 75.28±2.26 |
| TCP w/o $\mathcal{D}_t$ | 79.08±0.70 | 74.98±1.64 |

Table 3: Ablation study results on CIFAR10.

Figure 7: Transition matrix estimation errors on CIFAR10 from IDN-0.1 to IDN-0.4.

instance-dependent methods PTD (Xia et al., 2020b) and Bayes (Yang et al., 2021). As shown in Figure 7, our method achieves the consistent best estimation error on CIFAR10 with different noise rates.

**Ablation study.** We study the effect of removing different components of our methods to provide insights into what makes TCP successful in Table 3. TCP w/o $\mathcal{D}_c$ indicates that we do not select high clean-TCP data $\mathcal{D}_c$ to learn the clean classifier while TCP w/o $\mathcal{D}_t$ indicates that we do not select high noisy-TCP data $\mathcal{D}_t$ to learn the transition matrix and use it to fine-tune the clean classifier. Results show that the performances of both reduced methods decrease. Without $\mathcal{D}_c$, the primary clean classifier cannot be learned, and thus the transition matrix cannot be learned well. Without $\mathcal{D}_t$, the transition matrix is not learned, and thus the whole noisy data cannot be fully exploited to build the consistent classifier. To sum up, the learning of the clean classifier and the transition matrix benefit and boost each other.

## 6 CONCLUSIONS

In this paper, we study the instance-dependent label noise (IDN) problem, which is a more general and practical setting than the previously addressed instance-independent label noise problem. Targeting the main challenges, we propose a novel time-consistency metric, i.e., TCP for the IDN problem. Based on TCP, we can detect examples with clean labels or correct pseudo labels better than the existing measures, and allocate reliable triplets for learning the transition matrix. Then we design an assumption-free curriculum that learns the clean classifier, as well as the transition matrix simultaneously. Through extensive experiments, we empirically demonstrate that the proposed method remarkably surpasses the baselines on many datasets with both synthetic noise and real-world noise, and achieves the smallest transition matrix estimation error than existing methods.

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

# A    MORE EMPIRICAL STUDY REGARDING FIGURES 1 AND 2

## A.1    MORE EMPIRICAL STUDY REGARDING FIGURE. 1

Figures 8 and 9 show the results on more practical dataset CIFAR100 and IDN noise 0.4 higher IDN noise 0.6 with different partitions (2:6:2), which demonstrate that our claims hold in general and do not change sensitively with the partition ratios.

Figures 10 and 11 show the results on datasets CIFAR100 and SVHN with different model architectures. Conclusions from Section 3 are based on the memorization effect of overparameterized DNNs. Therefore, they hold better for deeper DNNs (ResNet50) than the shallower DNNs (AlexNet). Moreover, for AlexNet on SVHN, the high TCP group partitioned at an early stage has no overlap with the middle TCP group. Overall, the results demonstrate that our conclusions in the paper holds true and generalize to other architectures and datasets.

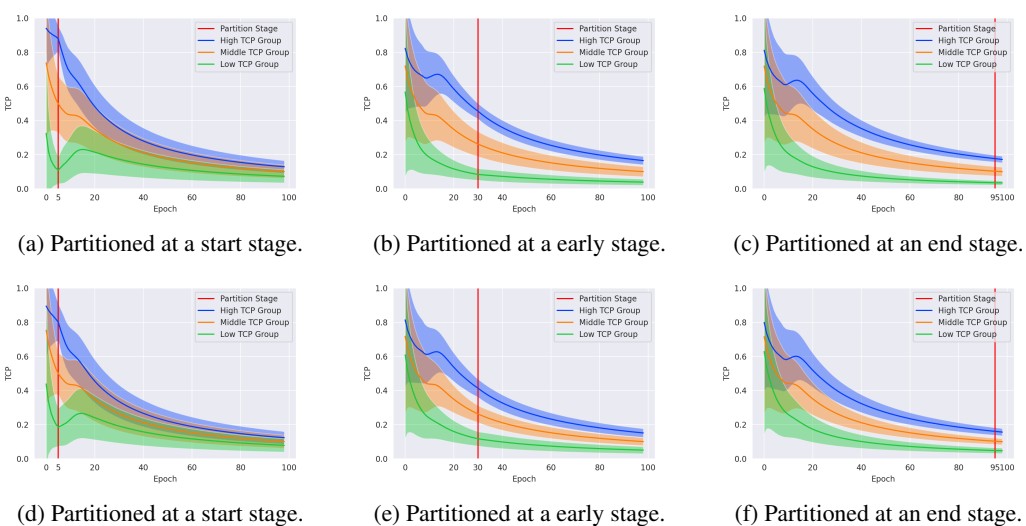

| (a) Partitioned at a start stage. | (b) Partitioned at a early stage. | (c) Partitioned at an end stage. |
| --- | --- | --- |
| (d) Partitioned at a start stage. | (e) Partitioned at a early stage. | (f) Partitioned at an end stage. |

Figure 8: TCP (mean and std.) of three groups partitioned by the TCP calculated at the start stage (epoch 5), early stage (epoch 30), and end stage (epoch 95) during training a ResNet50 on CIFAR100 with IDN-0.4 for 100 epochs. The first row is partitioned by high TCP (10%), middle TCP (80%), and low TCP (10%). The second row is partitioned by high TCP (20%), middle TCP (60%), and low TCP (20%).

## A.2    COMPARISON WITH SOTA SELECTION METHODS

Following the same setting as Figure 2, we select 5,000 confident examples at every epoch $t$ according to six types of selection criterion, i.e., instantaneous prediction $\text{InP}_t(x)$, instantaneous loss $\ell(x)$, time-consistency of prediction $\text{TCP}_t(x)$, time-consistency of loss, and two SOTA confident sample selection methods: FINE (Kim et al., 2021) and Topological Filter (Wu et al., 2020). Then we count the number of instances with clean labels and calculate the clean ratios. As shown in Figure 12, at the starting stage, when the model just learns the clean data while has not fit the noisy data, FINE and Topological Filter perform perform well. As the training goes and the model fits the noisy data, our method achieves the best selection clean ratio .

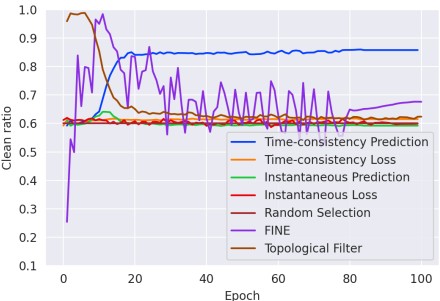

Figure 12:  Clean ratios of the selected top 5000 instances ranked by six kinds of instance hardness measures, respectively. The clean ratio of randomly selected instances is 0.6 since the noise rate is 0.4.

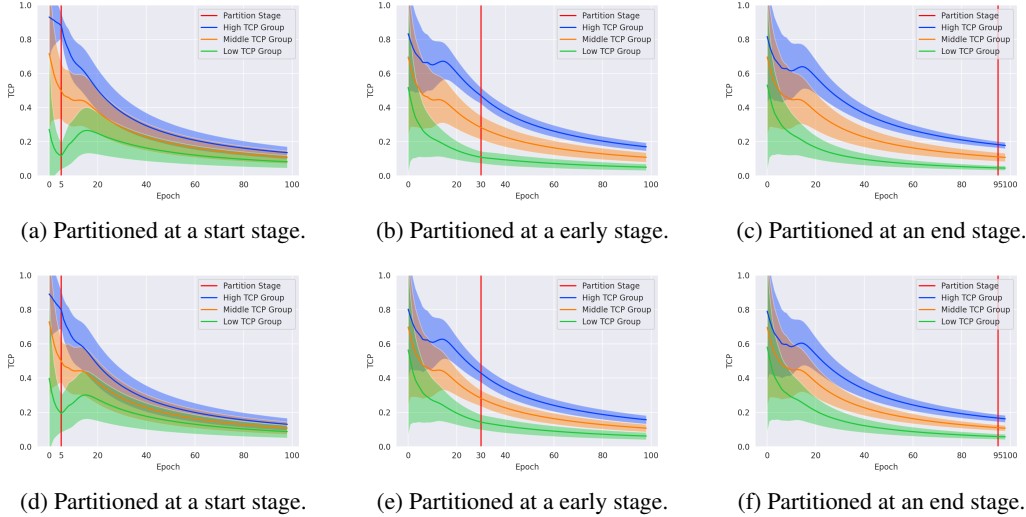

Figure 9: TCP (mean and std.) of three groups partitioned by the TCP calculated at the start stage (epoch 5), early stage (epoch 30), and end stage (epoch 95) during training a ResNet50 on CIFAR100 with IDN-0.6 for 100 epochs. The first row is partitioned by high TCP (10%), middle TCP (80%), and low TCP (10%). The second row is partitioned by high TCP (20%), middle TCP (60%), and low TCP (20%).

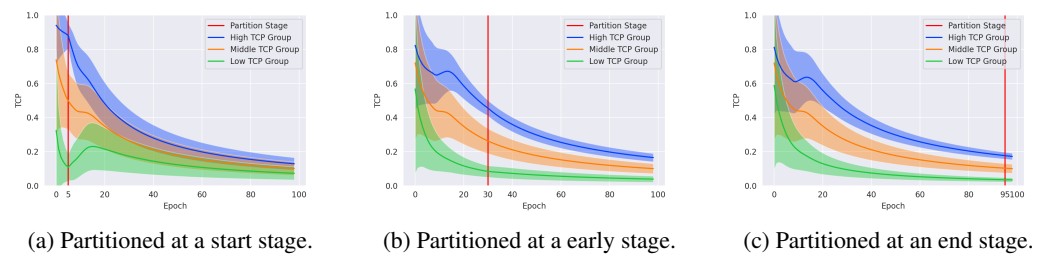

Figure 10: TCP (mean and std.) of three groups (high TCP (10%), middle TCP (80%), and low TCP (10%)) partitioned by the TCP calculated at the start stage (epoch 5), early stage (epoch 30), and end stage (epoch 95) during training a ResNet50 on CIFAR100 with IDN-0.4 for 100 epochs.

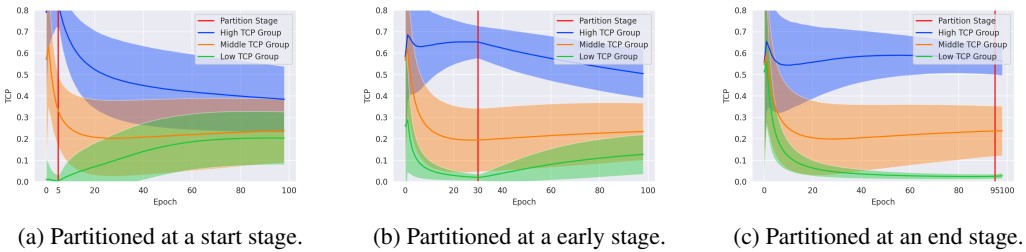

Figure 11: TCP (mean and std.) of three groups (high TCP (10%), middle TCP (80%), and low TCP (10%)) partitioned by the TCP calculated at the start stage (epoch 5), early stage (epoch 30), and end stage (epoch 95) during training a AlexNet on SVHN with IDN-0.4 for 100 epochs.

## B   THEORETICAL ANALYSIS ON THE EFFECT OF INTRODUCING INSTANCES WITH PSEUDO LABELS

Consider the situation we have a labeled set $\mathcal{L}$ (in practice it can be the selected confident examples set) and one unlabeled instance $x'$. By training on $\mathcal{L}$ for one step, we have

$$\theta_{t+1} = \theta_t + \eta \sum_{x \in \mathcal{L}} \nabla_\theta \ell (x; \theta_t),$$

and by training on $\mathcal{L}$ and $x'$ for one step, we have

$$\theta'_{t+1} = \theta_t + \eta \left( \sum_{x \in \mathcal{L}} \nabla_\theta \ell (x; \theta_t) + \nabla_\theta \ell (x'; \theta_t) \right),$$

where $\theta_t$ denotes the network parameters at step $t$ and $\eta$ denotes the learning rate.

The Taylor expansion of loss $\ell (x; \theta)$ at the point $\theta = \theta_0$ is:

$$g_{\theta_0} (\theta) = \left[ \sum_{x \in \mathcal{L}} \ell (x; \theta_0) + \nabla_\theta \ell (x; \theta_0) (\theta - \theta_0) \right] + o \left( (\theta - \theta_0)^2 \right). \tag{4}$$

Then we evaluate the forgetting effect of introducing instances $x'$ with its pseudo label to the training set by checking the change of loss over the labeled set $\mathcal{L}$ with $x'$ added or not. If adding $x'$ does not cause a vital change, we can conclude that it does not lead to catastrophic forgetting of the learned examples with correct labels. Therefore, we calculate the change of loss over the labeled set by

$$\frac{1}{\eta} \left| \sum_{x \in \mathcal{L}} \left[ \ell (x; \theta^{t+1}) - \ell \left( x; \hat{\theta}^{t+1} \right) \right] \right| = \frac{1}{\eta} \left| g_{\theta^t} (\theta^{t+1}) - g_{\theta^t} \left( \hat{\theta}^{t+1} \right) \right|$$

$$\approx \left| \nabla_\theta \ell (x'; \theta^t) \sum_{x \in \mathcal{L}} \nabla_\theta \ell (x; \theta^t) \right|$$

$$= \left| \frac{\partial \ell (x'; \theta^t)}{\partial \theta^t} \frac{\partial \theta^t}{\partial t} \right|$$

$$= \left| \frac{\partial \ell (x'; \theta^t)}{\partial t} \right|$$

$$= \left| \frac{\partial \ell (x'; \theta^t)}{\partial p_t^{\hat{y}'_t} (x')} \frac{\partial p_t^{\hat{y}'_t} (x')}{\partial t} \right|,$$

where $p_t^{\hat{y}'_t}(x')$ is the probability of $x'$ belonging to $\hat{y}'_t$ at step $t$, and $\hat{y}'_t$ is the prediction (pseudo label) of $x'$ at step $t$. The second line holds because we omit the second and higher order terms of the Taylor expansion in Eq equation 4. Then, with cross-entropy loss employed, we have

$$\frac{\partial \ell (x'; \theta^t)}{\partial p_t^{\hat{y}'_t} (x')} = \frac{\partial \log(p_t^{\hat{y}'_t} (x'))}{\partial p_t^{\hat{y}'_t} (x')} = -\frac{1}{p_t^{\hat{y}'_t} (x')}.$$

Next, by using $\left( p_{t+1}^{\hat{y}'_t}(x') - p_t^{\hat{y}'_t}(x') \right)$ to approximate $\frac{\partial p_t^{\hat{y}'_t}(x')}{\partial t}$, we have

$$\frac{1}{\eta} \left| \sum_{x \in \mathcal{L}} \left[ \ell (x; \theta_{t+1}) - \ell \left( x; \theta'_{t+1} \right) \right] \right| \approx \left| \frac{p_{t+1}^{\hat{y}'_t}(x')}{p_t^{\hat{y}'_t}(x')} - 1 \right|. \tag{5}$$

Since $x'$ is selected with high clean-TCP, $p_t^{\hat{y}'_t}(x')$ is very close to $p_{t+1}^{\hat{y}'_t}(x')$ because it has been verified in Figure 1 that instances with high clean-TCP in the early stage maintain their high clean-TCP in the future, which means the loss change can be bounded with a very small value. Therefore, exploiting high clean-TCP instances with pseudo labels helps to correct corrupted labels and learn a clean classifier without causing catastrophic forgetting of the learned examples with correct labels.

# C  DATA DISTRIBUTION IN TERMS OF CLEAN- AND NOISY-TCP AT DIFFERENT TRAINING STAGES

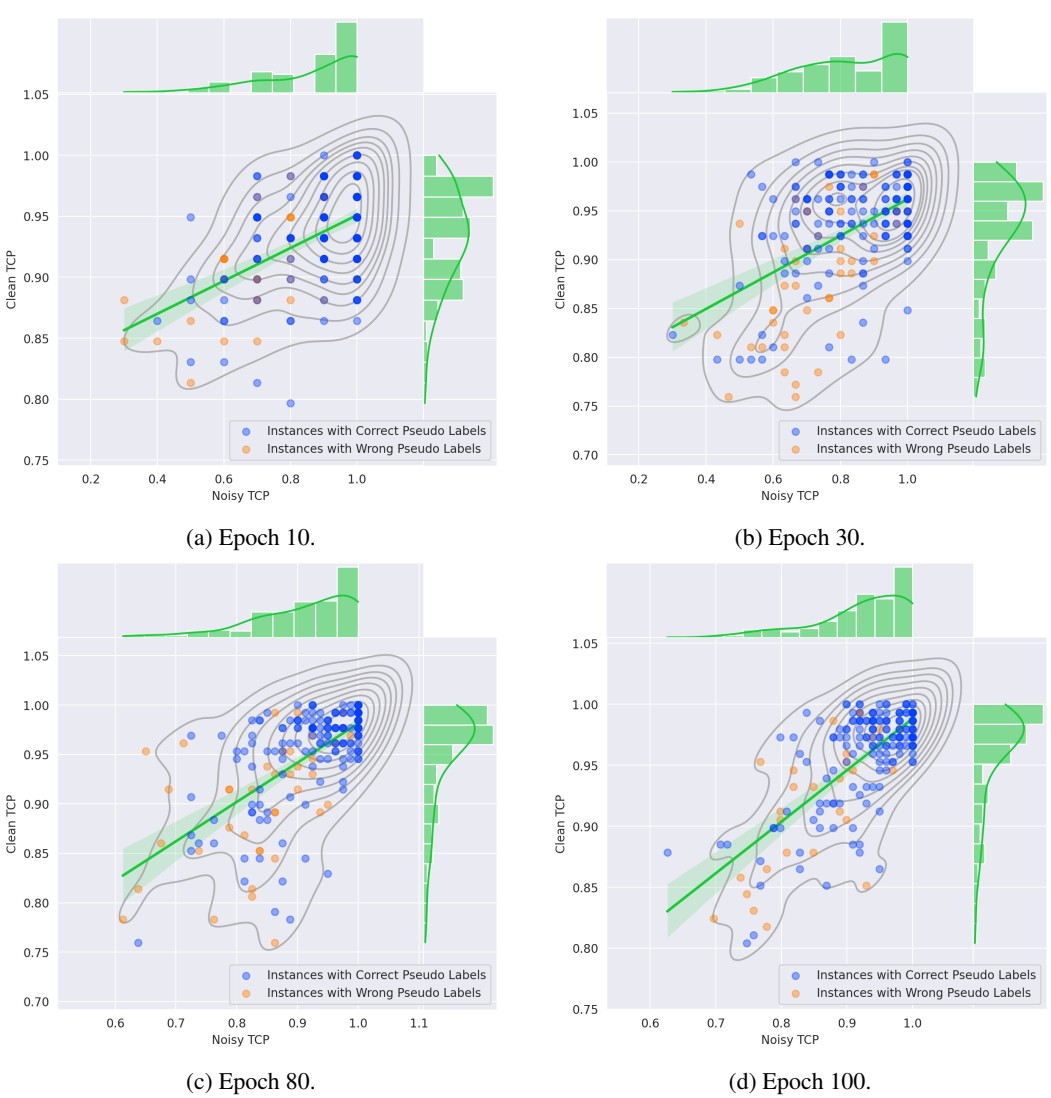

Figure 13: Clean ratios of selected high noisy-TCP examples w.r.t. their pseudo labels with exponential growth of the selected number and Spearman rank-order correlation coefficient between the noisy- and clean-TCP during our curriculum learning on CIFAR10 with IDN-0.4 for 100 epochs.

# D  ALGORITHM

# E  TRAINING DETAILS

We use a ResNet18 network for F-MNIST, a ResNet34 network for SVHN and CIFAR10, a ResNet50 network for CIFAR100, a PreAct-ResNet18 for CIFAR10N, and a pre-trained ResNet50 network for Clothing1M. The transition matrix is modeled by a linear layer which transforms the latent representation vector to a $c^2$ vector, and then reshaped to a $c \times c$ matrix. The batchsize is set to 32 for Clothing1M and 128 for others. The weight decay is set to $1e^{-4}$, $5e^{-4}$, 0, $5e^{-4}$, and 0.001 for F-MNIST, SVHN, CIFAR10/100, CIFAR10N, and Clothing1M, respectively. For synthetic noisy datasets, we use the Adam optimizer. At the warmup clean-TCP stage, the learning rate is initialized

---

**Algorithm 2** TCP Guided Curriculum Learning Algorithm.

---

**Input:** Noisy training sample $\mathcal{D}$, feature extractor $\phi$, classification layer $c$, transition matrix layer $t$, $f(\cdot) = c(\phi(\cdot))$, $T(\cdot) = t(\phi(\cdot))$, number sequences $N_c$ and $N_t$, training epoch $e_1$, $e_2$, and $e_3$.

1: **Warmup clean-TCP:**
2: **for** $e$ in $\{1, \cdots, e_1\}$ **do**
3:      Train $\phi$ and $c$ on $\mathcal{D}$ by minimizing a standard CE loss $\sum_{n=1}^{N} \mathcal{L}(f(x_n), \tilde{y}_n)$.
4:      Record the clean prediction and calculate the clean-TCP.
5: **end for**
6: **Warmup noisy-TCP:**
7: **for** $e$ in $\{1, \cdots, e_2\}$ **do**
8:      Select $N_c[e]$ high clean-TCP instances as $\mathcal{D}_c$.
9:      Train $\phi$, $c$ and $t$ on $\mathcal{D}_c$ by minimizing $\sum_{n=1}^{N} \mathcal{L}_n(f(x_n), \tilde{y}_n) + \sum_{n=1}^{N} \mathcal{L}(T^\top(x_n)f(x_n), \tilde{y}_n)$.
10:      Record the clean and noisy prediction and calculate the clean- and noisy-TCP.
11: **end for**
12: **Curriculum training:**
13: **for** $e$ in $\{1, \cdots, e_3\}$ **do**
14:      Select $N_t[e_2 + e]$ high noisy-TCP instances as $\mathcal{D}_t$.
15:      Train $t$ while fixing $\phi$ and $c$ on $\mathcal{D}_t$ by minimizing $\sum_{n=1}^{N} \mathcal{L}_n(T^\top(x_n)f(x_n), \tilde{y}_n)$.
16:      Select $N_c[e_2 + e]$ high clean-TCP instances as $\mathcal{D}_c$.
17:      Train $\phi$ and $c$ on $\mathcal{D}_c$ by minimizing $\sum_{n=1}^{N} \mathcal{L}_c(f(x_n), y_n^*)$, where $y^*$ is the pseudo label.
18:      Fix $t$ and fine-tune $\phi$ and $c$ on $\mathcal{D}$ by minimizing $\sum_{n=1}^{N} \mathcal{L}_n(T^\top(x_n)f(x_n), \tilde{y}_n)$.
19:      Record the clean and noisy prediction and calculate the clean- and noisy-TCP by Eq. equation 1.
20: **end for**

**Output:** Optimized feature extractor $\phi$, classification layer $c$, transition matrix layer $t$.

---

to 0.001 and decayed every 5 epochs with 50 epochs in total by a factor of 0.1, 1/3, and 1 for F-MNIST, SVHN, and CIFAR10/100, respectively. In the rest of 100-epoch training, the leaning rate of the feature extractor $\phi$ and classification layer $c$ is $1e^{-4}$ and divided by 10 at epoch 30 and 80; the leaning rate of the transition matrix layer $t$ is $3e^{-4}$ before epoch 30 and $1e^{-5}$ otherwise. The learning rate for fine-tuning is $1e^{-6}$. For real-world noisy dataset CIFAR10N and Clothing1M, we follow the optimization method as Dividemix. For CIFAR10N, in the warmup clean-TCP stage, the learning rate is initialized to 0.001 and decayed every 5 epochs with 50 epochs in total by a factor of 1/3. In the rest of 300-epoch training, the leaning rate of the transition matrix layer $t$ is $6e^{-3}$ before epoch 80 and $2e^{-4}$ between epoch 80 and 150, and $2e^{-4}$ otherwise. The learning rate for fine-tuning is $2e^{-3}$ before epoch 80, and $6e^{-4}$ between epoch 80 and 150, and $2e^{-4}$ between epoch 150 and 250, and $2e^{-5}$ otherwise. For Clothing1M, in the warmup clean-TCP stage, the learning rate is initialized to 0.002 and decayed every 2 epochs with 5 epochs in total by a factor of 1/3. In the rest of 20-epoch training, the leaning rate of the transition matrix layer $t$ is $6e^{-4}$ before epoch 8 and $2e^{-5}$ between epoch 8 and 12, and $5e^{-6}$ otherwise. The learning rate for fine-tuning is $2e^{-4}$ before epoch 10, and $6e^{-5}$ between epoch 10 and 14, and $2e^{-5}$ between epoch 14 and 17, and $2e^{-6}$ otherwise.

## E.1 TRAINING TIME

Due to the light-weight and simple network architecture, our method is more time-efficient and scalable than those methods, which employs dual networks or requires data augmentations for semi-supervised learning. We report the time costs below to demonstrate this advantage.

Table 4: The average time of training each component on CIFAR10 and CIFAR100 with ResNet34 on NVIDIA 3090.

|  | Standard training | Fine-tune | Estimating T |
|---|---|---|---|
| CIFAR10 | 38.42s | 46.19s | 1.34s |
| CIFAR100 | 37.37s | 47.06s | 1.57s |

The additional cost caused by estimating T and fine-tune is small. For each epoch, the additional time cost of estimating-T part is neglectable when compared with the cost of one standard training epoch minimizing the cross-entropy loss. This is because estimating T only updates the parameters of a $c \times c$ linear layer, where $c$ is the number of classes. The time cost of fine-tune part is slightly bigger than one standard training epoch. Fortunately, both parts are not necessary to be applied in every epoch. In our experiments, we only apply them at the last 50 epochs. Moreover, since traning clean classifier part (line 7 in Algorithm 1) and estimating T part only involves the high clean-TCP and high noisy-TCP data rather than the whole data, which save plenty of time for the fine-tune part. Therefore, in practice, our method can easily adapt and scale to meet the realistic settings.

## F  EXPERIMENT RESULTS

Table 5: Means and stds of classification accuracy on *F-MNIST* with different label noise rates.

|             | IDN-0.1        | IDN-0.2        | IDN-0.3        | IDN-0.4        | IDN-0.5        |
|-------------|----------------|----------------|----------------|----------------|----------------|
| CE          | 88.54±0.31     | 88.38±0.42     | 84.22±0.35     | 68.86±0.78     | 51.42±0.66     |
| Decoupling  | 89.27±0.31     | 86.50±0.35     | 85.33±0.47     | 78.54±0.53     | 57.32±2.11     |
| MentorNet   | 90.00±0.34     | 87.02±0.41     | 86.02±0.82     | 80.12±0.76     | 58.62±1.36     |
| Co-teaching | 90.82±0.33     | 87.89±0.41     | 86.88±0.32     | 82.78±0.95     | 63.22±1.58     |
| Co-teaching+| 90.92±0.51     | 89.77±0.45     | 88.52±0.45     | 83.57±1.77     | 59.32±2.77     |
| Joint       | 70.24±0.99     | 56.83±0.45     | 51.27±0.67     | 44.24±0.78     | 30.45±0.45     |
| DMI         | 91.98±0.62     | 90.33±0.21     | 84.81±0.44     | 69.01±1.87     | 51.64±1.78     |
| Forward     | 89.05±0.43     | 88.61±0.43     | 84.27±0.46     | 70.25±1.28     | 57.33±3.75     |
| Reweight    | 90.33±0.27     | 89.70±0.35     | 87.04±0.35     | 80.29±0.89     | 65.27±1.33     |
| T-Revision  | 91.56±0.31     | 90.68±0.66     | 89.46±0.45     | 84.01±1.24     | 68.99±1.04     |
| CRUST       | 89.53±0.55     | 89.20±0.58     | 86.68±0.92     | 83.48±1.55     | 69.59±3.60     |
| CausalINL   | 90.14±0.31     | 88.83±0.37     | 85.38±1.49     | 83.82±2.29     | 69.55±4.11     |
| PTD         | 91.01±0.22     | 90.03±0.32     | 87.68±0.42     | 84.03±0.52     | 72.43±1.76     |
| Bayes       | 92.01±0.22     | 91.42±0.71     | 89.64±0.41     | 81.21±1.13     | 74.62±2.47     |
| TCP         | **92.64±0.22** | **92.15±0.38** | **91.62±0.59** | **90.56±0.79** | **77.49±2.88** |

Table 6: Means and stds of classification accuracy on *SVHN* with different label noise rates.

|             | IDN-0.1        | IDN-0.2        | IDN-0.3        | IDN-0.4        | IDN-0.5        |
|-------------|----------------|----------------|----------------|----------------|----------------|
| CE          | 90.77±0.45     | 90.23±0.62     | 86.33±1.34     | 65.66±1.65     | 48.01±4.59     |
| Decoupling  | 90.49±0.15     | 90.47±0.66     | 85.27±0.34     | 82.57±1.45     | 42.56±2.79     |
| MentorNet   | 90.28±0.12     | 90.37±0.37     | 86.49±0.49     | 83.75±0.75     | 40.27±3.14     |
| Co-teaching | 91.33±0.31     | 90.56±0.67     | 88.93±0.78     | 85.47±0.64     | 45.90±2.31     |
| Co-teaching+| 93.05±1.20     | 91.05±0.82     | 85.33±2.71     | 57.24±3.77     | 42.56±3.65     |
| Joint       | 86.01±0.34     | 78.58±0.72     | 76.34±0.56     | 65.14±1.72     | 46.78±3.77     |
| DMI         | 93.51±1.09     | 93.22±0.62     | 91.78±1.54     | 69.34±2.45     | 48.93±2.34     |
| Forward     | 90.89±0.63     | 90.65±0.27     | 87.32±0.59     | 78.46±2.58     | 46.27±3.90     |
| Reweight    | 92.49±0.44     | 91.09±0.34     | 90.25±0.77     | 84.48±0.86     | 45.46±3.56     |
| T-Revision  | 94.24±0.53     | 94.00±0.88     | 93.01±0.83     | 88.63±1.37     | 49.02±4.33     |
| CRUST       | 93.22±1.32     | 91.55±0.36     | 88.64±1.43     | 80.75±2.78     | 58.30±2.77     |
| CausalINL   | 92.38±0.44     | 91.40±0.86     | 90.23±1.60     | 84.50±1.71     | 68.06±5.12     |
| PTD         | 93.21±0.45     | 92.36±0.68     | 90.57±0.42     | 86.78±0.63     | 55.88±3.73     |
| Bayes       | 94.71±0.44     | 94.02±1.32     | 91.38±1.94     | 85.55±3.17     | 75.46±3.79     |
| TCP         | **94.90±0.11** | **94.60±0.20** | **93.92±1.37** | **94.09±0.34** | **84.92±8.40** |

Table 7: Means and stds of classification accuracy on *CIFAR100* with different label noise rates. Note that PTD is not applicable to CIFAR100 which has large classes due to its matrix factorization component.

|  | IDN-0.1 | IDN-0.2 | IDN-0.3 | IDN-0.4 | IDN-0.5 |
|---|---|---|---|---|---|
| CE | 36.80±1.62 | 31.64±1.04 | 30.67±2.67 | 24.00±1.76 | 20.24±1.49 |
| Decoupling | 37.16±0.86 | 33.01±1.61 | 31.65±2.62 | 24.72±2.51 | 20.13±2.72 |
| MentorNet | 37.95±0.93 | 33.72±1.03 | 32.04±1.97 | 26.93±2.35 | 21.86±2.30 |
| Co-teaching | 38.57±0.95 | 35.60±1.49 | 33.77±1.91 | 26.17±2.35 | 21.96±2.51 |
| Co-teaching+ | 37.92±1.04 | 34.51±1.43 | 33.13±2.04 | 25.98±2.12 | 21.88±2.43 |
| Joint | 38.96±0.73 | 35.91±1.22 | 34.23±1.47 | 28.75±3.69 | 23.89±3.93 |
| DMI | 37.60±0.84 | 34.72±1.38 | 32.87±1.60 | 28.60±2.16 | 23.25±2.81 |
| Forward | 37.00±1.55 | 32.72±2.67 | 31.60±2.84 | 27.24±2.89 | 21.13±2.46 |
| Reweight | 37.11±0.98 | 33.98±1.68 | 32.60±1.22 | 27.83±1.27 | 22.01±3.26 |
| T-Revision | 38.03±1.05 | 34.42±2.32 | 33.60±1.98 | 28.15±3.69 | 22.12±3.67 |
| CRUST | 43.96±1.25 | 41.75±1.32 | 38.60±2.01 | 32.42±5.23 | 24.41±2.12 |
| CausalINL | 38.02±0.78 | 36.31±1.23 | 32.23±9.23 | 27.63±4.38 | 22.42±2.16 |
| Bayes | 40.76±1.98 | 36.56±1.20 | 29.26±1.67 | 24.38±1.39 | 17.66±0.94 |
| TCP | **49.65±0.43** | **46.28±2.56** | **44.12±1.92** | **39.88±0.62** | **29.45±2.35** |

