# OpenReview forum: "A Time-Consistency Curriculum for Learning from Instance-Dependent Noisy Labels"
_ICLR.cc/2023/Conference — Submitted to ICLR 2023_

### Official Review · Reviewer_vCae · 2022-10-21

**Confidence:** 4
**Correctness:** 3
**Technical Novelty And Significance:** 2
**Empirical Novelty And Significance:** 2
**Recommendation:** 3

**Clarity, Quality, Novelty And Reproducibility:**

This work is overall clear and easy to follow, but some definitions are missing. The proposed method is inherently similar to thresholding-based pseudo-labeling/correction methods in noisy label learning [1] and the originality is doubtful.

**Strength And Weaknesses:**

Pros:

1. This work provides a different perspective on selecting clean examples for noisy-label learning by the consistency of predictions, which brings some insights to the community.

2. The authors provided thorough analytical results of the TCP scores.

Cons:

1. My first concern is that the proposed TCP score, while seemingly new, may be analogous to the pseudo-labeling procedure of semi-supervised learning (SSL) and inherently not a new method. In SSL, methods like FixMatch clip the model prediction on unlabeled data as their pseudo-labels for training; Pi-Model/Mean-Teacher adopt a time-consistent prediction as pseudo-labels. When the model is confident to one instance (for example, max prediction > 0.95), its prediction would also be stable as well and has TCP scores. Such a pseudo-labeling method had also adopted in the noisy-label learning regime [1]. In fact, I doubt TCP is inherently a semi-supervised learning algorithm and may not be better than such a pseudo-labeling strategy.

2. The relationship between the TCP score and the instance-dependent noise is unclear. Why is TCP particularly important to the ID setup? In effect, as the proposed method also integrates the SSL techniques, I believe simple pseudo-labeling can also accomplish the anchor sample selection procedure for transition matrix estimation.

3. The empirical study is weak for me.

    3.1 First, the real-world noisy label CIFAR dataset has a version of CIFAR-100-N, but the results are reported in neither Table 2 nor Appendix. Does the proposed method underperform on this dataset?

    3.2 Some important baselines are missing. For example, the UNICON, and SOP methods have shown promising results. The seminal DivideMix method is not compared in Table 1. While it's not mainly designed for ID NLL, I believe it can also handle this task empirically. Notably, TCP might be a semi-supervised NLL method and simply comparing it with naive NLL baselines is unfair.

    3.3 Second, the ablation study is really weak. The authors should thoroughly analyze the effectiveness of different components. Does the estimated transition matrix really work? What if we directly run a dividemix model with TCP-selected samples? Can we replace the TCP-based selection with a pseudo-label-based selection for the transition matrix?

    3.4 The ablation is mainly conducted on CIFAR-10, which is not convincing.

4. I didn't find the definition of \hat{y}, which made me hard to understand Eq. (1).

[1] Li J, Xiong C, Hoi S. MoPro: Webly Supervised Learning with Momentum Prototypes[C]//International Conference on Learning Representations. 2020.

[2] Karim N, Rizve M N, Rahnavard N, et al. UNICON: Combating Label Noise Through Uniform Selection and Contrastive Learning[C]//Proceedings of the IEEE/CVF Conference on Computer Vision and Pattern Recognition. 2022: 9676-9686.

[3] Liu S, Zhu Z, Qu Q, et al. Robust Training under Label Noise by Over-parameterization[J]. ICML, 2022.

**Summary Of The Paper:**

This work targets to the instance-dependent noisy-label learning problem and proposes a new curriculum learning-based algorithm. In particular, the authors calculate the 'time-consistency of prediction' (TCP) scores that indicate how consistent are the predictions of samples over the course of training. Then, the proposed framework selects anchor instances by the TCP scores and estimates the transition matrices. Experiments show the proposed method achieves better performance than the baselines.

**Summary Of The Review:**

The originality is doubtful and the empirical study is weak. So I vote for rejection.

---

> ### Author Response · Authors · 2022-11-19
> **Response**
>
> # Reviewer vCae
> Thank you for your constructive comments! Below we carefully address your concerns one by one.
>
> Q1: I doubt TCP is inherently a semi-supervised learning algorithm and may not be better than such a pseudo-labeling strategy.
> -----------------
> **A1:** TCP is metric instead of a pseudo-labelning strategy. In this work, we use it to detect examples with clean labels or correct pseudo labels better than
> the existing measures, and allocate reliable triplets for learning the transition matrix.
>
>
> Q2: Why is TCP particularly important to the ID setup? In effect, as the proposed method also integrates the SSL techniques, I believe simple pseudo-labeling can also accomplish the anchor sample selection procedure for transition matrix estimation.
> -----------------
> **A2:** In instance-dependent label noise case, there are two main challenges (1) the "small loss" trick is no longer effective in detecting correct labels because the loss threshold drastically varies across instances and is determined by each transition matrix $T(X)$; (2) the instance-dependent transition matrix $T(X)$ is not identifiable given only the noisy sample and it heavily relies on the estimation of clean label $Y$ in the triple $(X,Y,\tilde{Y})$, which is an unsolved challenge in (1).
>
> TCP is particularly important to the ID setup as it can detect examples with clean labels or correct pseudo labels better than the existing measures, and allocate reliable triplets for learning the transition matrix.
>
> Q3: The real-world noisy label CIFAR dataset has a version of CIFAR-100-N, but the results are reported in neither Table 2 nor Appendix. Some important baselines are missing. For example, the UNICON, and SOP methods have shown promising results.
> -----------------
> **A3:** We added the results of a new dataset CIFAR100N and a new method SOP to the table of real-world noisy datasets results, which is shown below. Our method achieves compelling performance.
> | | CIFAR10N-1 | CIFAR10N-2 | CIFAR10N-3 | CIFAR10N-W | CIFAR100N | Clothing1M    |
> |  ----  | ----  | ---- |  ----  | ----  | ---- | ---- |
> PES (semi) | 95.06$\pm$0.15 | 95.19$\pm$0.23  | 95.22$\pm$0.13	 | 92.68$\pm$0.22 | 70.36$\pm$0.33 | 74.29|
> DivideMix | 95.16$\pm$0.19 | 95.23$\pm$0.07 | 95.21$\pm$0.14	 | 92.56$\pm$0.42 | **71.13$\pm$0.48** | 74.30	|
> CORES | 94.45$\pm$0.14 | 94.88$\pm$0.31	 | 94.74$\pm$0.03	 | 91.66$\pm$0.09 | 61.15$\pm$0.73 | 73.24		 |
> ELR+ | 94.43$\pm$0.41 | 94.20$\pm$0.24	 | 94.34$\pm$0.22	 | 91.09$\pm$1.60 | 66.72$\pm$0.07 | 74.31	|
> JoCoR | 90.30$\pm$0.20 | 90.21$\pm$0.19	 | 90.11$\pm$0.21	 | 83.37$\pm$0.30 | 59.97$\pm$0.24 | 70.30	 |
> CAL | 90.93$\pm$0.31 | 90.75$\pm$0.30	 | 90.74$\pm$0.24	 | 85.36$\pm$0.16 | 61.73$\pm$0.42 | 74.21	|
> **SOP** | 95.28$\pm$0.13 | 95.31$\pm$0.10	 | 95.39$\pm$0.11	 | 93.24$\pm$0.21 | 67.81$\pm$0.23 | 73.50	|
> TCP-D | **95.51$\pm$0.06** | **95.37$\pm$0.08** | **95.43$\pm$0.04** | **93.36$\pm$0.09** | 70.09$\pm$0.13 | **74.41**|
>
> Q4: The ablation study is really weak. Does the estimated transition matrix really work? What if we directly run a dividemix model with TCP-selected samples? Can we replace the TCP-based selection with a pseudo-label-based selection for the transition matrix?
> -----------------
> **A4:** TCP-D is the method that additionally applies our transition matrix learning and fine-tuning parts to the SoTA method Dividemix. Note that fine-tuning part is attached to the transition matrix learning part otherwise the estimated T is not involved. Then, in light of the improvement of TCP-D over Dividemix on CIFAR10N and Clothing1M datasets, which is exactly an ablation study on the estimated transition matrix, the estimated transition matrix really works.
>
> Dividemix is a systematic work, and the components in it are entangled with each other. Therefore, directly (naively) running a dividemix model with TCP-selected samples is not a good idea, which requires lots of effort in tuning the hyper-parameters, e.g., Dividemix uses different sample selection thresholds for different settings.
>
> I don't get the idea of replacing the TCP-based selection with a pseudo-label-based selection for the transition matrix. To select the high-quality triplets $(X,Y,\tilde{Y})$ for learning the transition matrix, two conditions should be considered. First, it is necessary for $f(\cdot)$ to output a precise clean class posterior, otherwise, $T$ cannot be optimized in the correct direction. Second, the noisy-TCP should be high. Therefore, high noisy-TCP instances not only are naturally stable for the new objective to predict $\tilde{y}$ but also satisfy the necessary condition to have precise clean class posterior, which makes them perfect examples for learning the transition matrix.
>
> Q5: The definition of \hat{y}.
> ------------------
> **A5:** \hat{y} is the predicted label.

---

> > ### Comment · Reviewer_vCae · 2022-11-22
> > **Response**
> >
> > > TCP is metric instead of a pseudo-labelning strategy.
> >
> > As I have mentioned in my comments, a high-TCP score data point is possibly a *high-prediction-score* data example. This makes the proposed method a semi-supervised learning technique when performing data selection with TCP scores. This procedure is very like the pseudo-labeling method of FixMatch.
> >
> > > Our method achieves compelling performance.
> >
> > As a result, it would be interesting to see whether TCP score is better than other SSL-based methods instead of usual sample selection methods. According to the newly-posted experimental results, I don't think TCP-D obtains statistically significant performance improvement.
> >
> > >I don't get the idea of replacing the TCP-based selection.
> >
> > I mean a new variant of Dividmix with TCP-based selection to see whether TCP selection is better than GMM-based selection.
> >
> > Moreover, I noticed that other reviewers also raised concerns about the experimental results (eg, Q4 of zfb4), but the authors didn't properly answer.

---

> > > ### Author Response · Authors · 2022-11-24
> > > **Response**
> > >
> > > > a high-TCP score data point is possibly a high-prediction-score data example. This makes the proposed method a semi-supervised learning technique when performing data selection with TCP scores.
> > >
> > > What is your concern about the possible relationship between the TCP score and the prediction-score?
> > >
> > >
> > > > it would be interesting to see whether TCP score is better than other SSL-based methods instead of usual sample selection methods.
> > >
> > > DivideMix and PES (semi) are SSL-based methods, which have been compared. To the best of my knowledge, the SSL technique is usually combined with the sample select technique in the noisy label learning community. Can you elaborate on what kind of SSL-based methods (instead of usual sample selection methods), that is proposed to deal with noisy label problem, you are interested in?
> > >
> > >
> > > > I don't think TCP-D obtains statistically significant performance improvement.
> > >
> > > Among these SoTA methods, which one do you think obtains statistically significant performance improvement? Which one do you think achieves more compelling performance?
> > >
> > >
> > > > Whether TCP selection is better than GMM-based selection.
> > >
> > > As I explained, dividemix is a systematic work, and the components in it are entangled with each other. Therefore, directly (naively) running a dividemix model with TCP-selected samples is not a good idea, which requires lots of effort in tuning the hyper-parameters, e.g., Dividemix uses different sample selection thresholds for different settings.
> > >
> > > Therefore, for fair comparison, we adopt the framework in Figure 2 in our paper, i.e., select confident examples at every epoch t with a fixed number 5,000 and check the clean ratio. Note that the GMM model is learned from the loss, which means the order of the probability obtained from the GMM model is same as the reverse order of the loss, i.e., instance of the lowest loss has the highest probability to be confident. Then the GMM-based selection is equivalent to the instantaneous-loss-based one, which is no better than the TCP selection.

---

> > > ### Author Response · Authors · 2022-11-29
> > > **Sincerely looking forward to your reply.**
> > >
> > > We appreciate your constructive comments on our paper. We hope we have the opportunity to see your further comments, answer any additional questions, and ultimately improve the quality of our submission. Are there unclear explanations? We could further clarify them.
> > >
> > > Best wishes, Authors

---

### Official Review · Reviewer_GBT4 · 2022-10-24

**Confidence:** 4
**Correctness:** 3
**Technical Novelty And Significance:** 2
**Empirical Novelty And Significance:** 3
**Recommendation:** 6

**Clarity, Quality, Novelty And Reproducibility:**

Clarity - Overall the paper could use major revisions to organization of the paper as well as a few passes to correct grammatical errors.  Because of the shortcomings in clarity, the paper is not easy to read despite the techniques being relatively simple.

Quality - The authors make a strong attempt at providing intuition and empirical justification for their proposed method.  They also review a large amount of prior work. Presentation, organization, and clarity issues aside, I believe the work is of moderate to high quality.

Novelty - The main novel contribution is the TCP heuristic used to select clean and noisy instances.  The heuristic is empirically argued for, but no strong theoretical analysis is provided.  As such, the paper, while empirically successful, is not very novel.

Reproducibility - Mainly because a number of baselines are not fully discussed nor is code provided, I think it would be rather difficult to reproduce the results of this work.

**Strength And Weaknesses:**

Strengths
1. Clearly techniques that learn from instance-dependent noise have considerable practical applications.
2. Their approach is relatively simple and seems like it could be implemented easily with common machine learning packages.
3. In terms of classification accuracy, the proposed approach achieves impressive performance relative to baselines.

Weaknesses
1. The paper suffers from poor organization which makes its somewhat difficult to follow.  Namely, there needs to be a self-contained "preliminaries" before describing the techniques that are introduced.  As it is written right now, lots of notation and leveraged prior work are introduced throughout the paper.  As a result, it is both difficult to follow the notation and difficult to identify what specifically was novel about the proposed techniques.  I recommend taking a step back and thinking about how to reorganize the paper entirely, and also consider discussing the most related IDN label learning algorithms in a more formal way to establish them as a point of reference to the proposed new contributions made by this work.
2. The claim of the proposed technique being immune to catastrophic forgetting is discussed as a formal contribution, but it neither formally defined nor strongly argued.  There needs to be a formal statement of what is being proved.  Also, the proof hinges on the statement: "Since x′ is selected with high clean-TCP... has been verified in Figure 1".  This is not a formally proved statement, which makes the "proof" more of an empirical observation.
3. The curriculum learning algorithm requires training two models to completion before training begins.  It is not clear how computationally expensive this is relative to the baselines.  To provide a more fair understanding of the performance of the proposed technique, I think wall run times should be provided for the training of the models in the empirical results section.


**Summary Of The Paper:**

The authors propose a method for training a classifier on data with Instance-Dependent Noisy Labels (IDN Labels, i.e. where label noise is a function of the instances themselves).  Their technique is within a class of techniques that monitor a heuristic during training that measures some notion of noise in labels, and then using instances selected as "clean" and "noisy" then learn a transformation matrix that describes the relationship between true and noisy labels in an instance-dependent manner.  The main novel observation is the the heuristics in prior work are sensitive enough that monitoring them from one training epoch to another doesn't provide clear signal  as to which instances and "clean".  They instead use a TCP metric that is a running average of whether the predicted label has changed between gradient updates.  They show that if you assume that their metric is relatively stable, then their curriculum learning approach that introduces new instances during training does not result in catastrophic forgetting.  Empirically, the proposed method outperforms a number of baselines, both with real and synthetic noise.

**Summary Of The Review:**

While the clarity of the paper is a barrier for understanding and the proposed technique is not of significant novelty, the authors provide strong intuition for their approach and follow that with strong empirical performance.  Because this technique seems to be the empirical leader for the IDN Labels learning problem, I am leaning towards acceptance.  I do think for a camera ready copy, the authors should take a fine-grained critical look at the writing and organization of the paper to ensure that the main contributions are more clearly articulated.

---

> ### Author Response · Authors · 2022-11-19
> **Response**
>
> # Reviewer GBT4
> Thank you for your constructive comments! Below we carefully address your concerns one by one.
>
> Q1: Poor organization.
> -----------------
> **A1:** We have refined our paper in the newly uploaded version.
>
>
> Q2: Formal statement of catastrophic forgetting.
> -----------------
> **A2:** We theoretically prove that additionally introducing a high clean-TCP instance with its pseudo label only contributes a bounded small change to the loss, which relieves the model of catastrophic forgetting of the learned confident examples.
>
> Q3: The curriculum learning algorithm requires training two models to completion before training begins. It is not clear how computationally expensive this is relative to the baselines.
> -----------------
> **A3:** We only need one model for TCP. As described in Algorithm 1, the model contains a feature extractor $\phi$, a classification layer $c$, and a transition matrix layer $t$. The standard clean classifier is $f(\cdot) = c(\phi(\cdot))$, while the noisy classifier is the $T(\cdot)f(\cdot)$, where $T(\cdot) = t(\phi(\cdot))$. Note that in the warmup stage, the $T-$part is not updated, which means it is equivalent to a widely adopted standard warmup on the noisy data.

---

> > ### Comment · Reviewer_GBT4 · 2022-11-22
> > **Response to Rebuttal**
> >
> > Thank you for your rebuttal.  I have read it, and have decided to keep my score at a 6.

---

### Official Review · Reviewer_zfb4 · 2022-10-24

**Confidence:** 4
**Correctness:** 3
**Technical Novelty And Significance:** 3
**Empirical Novelty And Significance:** 3
**Recommendation:** 5

**Clarity, Quality, Novelty And Reproducibility:**

The paper discusses a novel and empirically strong metric for curriculum learning that challenges the widely used small-loss assumption. Many readers in the field should find the work interesting. The paper is well written.

**Strength And Weaknesses:**

Strength:

1. The proposed metric TCP for curriculum learning is theoretically motivated. In particular, empirical results show that it is a very interesting alternative that outperforms several existing loss based selection criteria.
2. Integrating the TCP, Algorithm 1 is a nice general framework to combine both curriculum learning and label noise transition learning. Empirical experiments are quite strong.

Weaknesses / Major Concerns:

1. Regarding the motivation of TCP, in particular TCP vs. time consistency of loss (TCL), I find it a bit surprising that TCL works so much worse than TCP (Fig 2). The authors explained that 1-hot prediction is more robust than loss at instance level (second paragraph of Sec 3), but the authors later also said clean-TCP instances' loss change can be bounded with a very small value (left to Fig 3). May the authors give more justifications?
2. When motivating the curriculum learning in Sec 4, Fig 3 sees a drastically decreasing clean ratio of original noisy labels, while Fig 2 sees TCP clean ratio still stay high towards the end of training. Why is this? Is this due to the linearly increasing curriculum size in Fig 3? There is some prior work that shows fixing the number of instances being selected given small losses is probably not a good idea (e.g. https://arxiv.org/pdf/2104.02570.pdf), should consider setting a (dynamic) loss threshold instead.
3. Continuing on the previous question, how is the number of selected instances decided at each epoch in the experiments (e.g. linear/exponential growth as done in Fig 3/4)? Is there a way to dynamically control the number of selected instances with TCP?
4. Despite that the synthetic data experiments look very promising, the authors decide to use a variant of TCP-D in the real world data experiments (Tab 2). TCP-D extends DivideMix by selecting high noisy-TCP data to learn the transition matrix and using it to fine-tune the whole data. Given my understanding of the contribution of the paper, would it make more sense to actually replace the GMM-based sampling selection part of DivideMix by TCP guided curriculum? Why the particular choice of TCP-D? Did the authors try other extensions of DivideMix inspired by TCP?

Minor comments:

1. Should the authors include EMA over loss (Zhou et al. 2020) as it has been shown to be more robust than instantaneous loss, since TCP also considers the entire training history?
2. I think TCP w/o D_t in Tab 3 performs well enough to be standalone methods given that they are much simpler alternatives to the full model without the whole label transition or extra fine tuning on entire data (pending more results?)

**Summary Of The Paper:**

The paper proposes a novel metric, Time Consistency Prediction (TCP), as criteria to guide curriculum learning for jointly training the label transition matrix and clean classifier.

**Summary Of The Review:**

I believe the novelty and contribution of the paper outweigh the weaknesses of current theories, results and writing. I'm happy to increase my score if the authors can address some of my concerns.

---

> ### Author Response · Authors · 2022-11-19
> **Response**
>
> # Reviewer zfb4
>
> Thank you for your constructive comments! Below we carefully address your concerns one by one.
>
>
> Q1: I find it a bit surprising that TCL works so much worse than TCP (Fig 2). The authors explained that 1-hot prediction is more robust than the loss at the instance level (second paragraph of Sec 3), but the authors later also said clean-TCP instances' loss change can be bounded with a very small value (left to Fig 3). May the authors give more justifications?.
> -----------------
> **A1:** In Figure 2, TCP performs better than TCL on **selecting instances with clean labels**.
> In Eq.(2), we show that additionally introducing **high clean-TCP instance with its pseudo label** contributes a bounded small change to the loss, and thereby the gradient, and thereby the parameter of the DNN, which does not cause catastrophic forgetting of the learned confident examples.
>
>
> Q2: Fig 3 sees a drastically decreasing clean ratio of original noisy labels, while Fig 2 sees TCP clean ratio still stay high towards the end of training. Why is this? There is some prior work that shows fixing the number of instances being selected given small losses is probably not a good idea, should consider setting a (dynamic) loss threshold instead.
> -----------------
> **A2:** First, the training settings are different. It was vanilla training in Figure 2, and curriculum training in Figure 3. Second, the sample size changes.
> Setting a (dynamic) loss threshold could be a promising direction, but is out of the scope of this paper.
>
>
> Q3: How is the number of selected instances decided at each epoch in the experiments? Is there a way to dynamically control the number of selected instances with TCP?
> -----------------
> **A3:** For the number of selected high clean-TCP instances, the starting number is 2,500 (clean_start_num=2500); the vanilla training classification accuracy on the noisy dataset (obtained from the warmup state) is regarded as the clean rate of the dataset, and the ending number (clean_select_max_num) is the estimated number of clean instances in the dataset, which is calculated as the total number of the dataset times the clean rate. Then the number growth function is np.linspace(clean_start_num, clean_select_max_num, end_num_epoch, endpoint=True, dtype=int), where the end_num_epoch is set to 95 in a 100-epoch training. For the number of selected high noisy-TCP instances, the number growth function is np.logspace(start=50, stop=100, end_num_epoch, endpoint=True, base=1.1131, dtype=int), which generates a sequence starts at base ** start and ends with base ** stop. The 50, 100, and 1.1131 are set to make the sequence starts with a small number (1.1131^50≈212) and ends with the training data size (1.1131^100≈45021). The end_num_epoch is set to 95 in a 100-epoch training.
>
> Dynamically (adaptively) controlling the number of selected instances will be investigated in our future work.
>
>
> Q4: Would it make more sense to actually replace the GMM-based sampling selection part of DivideMix by TCP guided curriculum? Why the particular choice of TCP-D? Did the authors try other extensions of DivideMix inspired by TCP?.
> -----------------
> **A4:** Using TCP-guided curriculum to estimate the transition matrix is our main contribution which is not overlapped with the components in Dividemix. Therefore, we choose to apply our transition matrix learning and fine-tuning parts to the SoTA method Dividemix.
>
>
> Q5: Should the authors include EMA over loss (Zhou et al. 2020) as it has been shown to be more robust than instantaneous loss?
> ------------------
> **A5:** An exponential moving average (EMA) at step t+1 is recursively defined as $EMA_{t+1} = \lambda a_t + (1 - \lambda)EMA_{t}$, where $EMA_{t}$ is the EMA at step $t$, $a_t$ is the current statistic, and $\lambda$ is a discount factor.
>
> Hence, the time-consistent prediction/loss is exactly an EMA of prediction/loss with a discount factor $\lambda=1/(t+1)$ at step $t$, which works promisingly in experiments. We will investigate how to find an adaptive discount factor for different label noise ratios in our future work.

---

> ### Author Response · Authors · 2022-11-29
> **Sincerely looking forward to your reply.**
>
> We appreciate your constructive comments on our paper. We hope we have the opportunity to see your further comments, answer any additional questions, and ultimately improve the quality of our submission. Are there unclear explanations? We could further clarify them.
>
> Best wishes,
> Authors

---

### Official Review · Reviewer_FeeE · 2022-10-27

**Confidence:** 4
**Correctness:** 2
**Technical Novelty And Significance:** 2
**Empirical Novelty And Significance:** 2
**Recommendation:** 3

**Clarity, Quality, Novelty And Reproducibility:**

As mentioned in detail above, the clarity and quality of this paper may be questionable. The novelty may be limited.

**Strength And Weaknesses:**

Strength:
* The paper is overall organized well
* Introduction to the related work is thorough and informative
* Learning the transition matrix and clean classifier simultaneously seems to be an interesting idea

Weaknesses:


**Clarity:** I believe overall this paper is not well-clarified, which greatly hinders reading and comprehension. I'll list some of the clarity issues below for reference.
* First, the background section is not sufficient in introducing the terms. For example, terms like "pseudo label", "catastrophic forgetting", "corrupted labels" in section 4.1 are never defined. I am having a hard time understanding this section without a clear definition of these terms. Most of the time I have to guess their meanings based on my own background knowledge.
* What does Equation (2) mean? I guess the point here is to prove the change of loss is small for high-TCP examples, but why this is important? I think the proposed metric is not based on losses.
* Quote, "instances with clean labels are mutually consistent with each other in producing gradient updates". This sentence is frequently mentioned in the paper, e.g. Section 3, 4.1, and thus I believe it should be important. But what does this mean? I am expecting some formula to clarify this but there is no. I might miss something here.
* Figure 2 shows the proposed metric is good at identifying correct labels when there is label noise. But how does this label noise generated? The authors only mentioned "manually add IDN at 0.4 noise rate (IDN-0.4) onto a benchmark dataset CIFAR10", but I didn't find any details of this, even in the appendix. I think such detail is critical here, as we would expect the proposed metric not to specialize on some particular type of label noise. Experiments on most label noise settings here can also be helpful but I didn't find any figures similar to Figure 2, but experimenting on different label noise settings.

**Quality**: There are quite a few unjustified premises in this paper, which cause difficulty for one to judge the contribution. For example,
* In section 3, quote, "Apparently, at the instance level, the one-hot prediction of an instance is a more robust metric than the loss because the former has a tolerance to the change of predicted class posterior while the latter has not". But this doesn't appear apparent to me. I can also justify an opposite claim in a similar way, namely the loss considering the entire probabilistic prediction thus is more robust.
* In Section 4.2, quote, "high noisy-TCP instances naturally indicate the instance is learned better and faster for predicting y, leading to stable and fast learning.". But I cannot see the reason here. Why is this premise natural?
* I also have some questions about the experiment results. In Table 1, why baselines such as Dividemix are missing? Is it because Dividemix is not applicable here? In Table 2, why a new method is introduced (TCP-D)? What about the performance of the original method? And there are very few descriptions of this new method.

**Novelty**:
* I believe the proposed metric itself is not novel. Using prediction stability to identify correct or noisy labels is seen in previous works, e.g. [1].
* The idea that using a designated model to learn the transition matrix is relatively novel to me. But apart from one sentence in the abstract, I didn't find any further explanation or justification for this idea.

[1] An Empirical Study Of Example Forgetting During Deep Neural Network Learning. Toneva et al.




**Summary Of The Paper:**

This paper studies robust learning on instance-dependent label noise. It proposes to discover the noise transition matrix and identify clean labels at the same time during training. It also proposes a metric to measure the probability of a label being correct, which is based on historical predictions of an instance during training. Such a metric can be used to better identify both correct labels and noisy labels, which facilitates the learning of noise transition matrix and clean labels.

**Summary Of The Review:**

I believe there are some interesting ideas in this paper. But right now, the clarity issue makes it almost impossible to judge the novelty and contribution. I think significant revision is required to improve the paper.

---

> ### Author Response · Authors · 2022-11-19
> **Response**
>
> # To Reviewer FeeE
> Thank you for your constructive comments! Below we carefully address your concerns one by one.
>
> Q1: The background section is not sufficient in introducing the terms， e.g., "pseudo label", "catastrophic forgetting", and "corrupted labels".
> -----------------
> **A1:** "pseudo label" and "corrupted labels" are plain language.
> "pseudo label" denotes **the label assigned by the classifier** as explained in the paper: "high clean-TCP indicates the **pseudo labels assigned by the classifier** are correct".
>
> "corrupted labels" denotes the label that is corrupted.
>
> "catastrophic forgetting" denotes the tendency of DNNs to forget previously learned information upon learning new information as explained in the paper: **"cause catastrophic forgetting of the learned confident examples."**
>
>
>
> Q2: What does Equation (2) mean? Why is this important?
> -----------------
> **A2:** Equation (2) indicates that additionally introducing a high clean-TCP instance with its pseudo label contributes a bounded small change to the loss, and thereby the gradient, and thereby the parameter of the DNN, which does not cause catastrophic forgetting of the learned confident examples.
>
> It is important because if the new data is not properly selected, it will cause catastrophic forgetting as the DNN forgets previously learned information upon learning new information.
>
>
> Q3: What does "instances with clean labels are mutually consistent with each other in producing gradient updates" mean?
> -----------------
> **A3:** Given two positive data points $\{x_{p1}, x_{p2}\}$ and two negative data points $\{x_{n1}, x_{n2}\}$. The set of instances with clean labels $\{(x_{p1}, 1), (x_{n1}, -1)\}$ produces gradient updates to guide the classifier to assign positive labels to positive instances and negative labels to negative instances, i.e., instances with clean labels are mutually consistent with each other in producing gradient updates. In contrast, the set of instances with noisy labels $\{(x_{p1}, 1), (x_{n1}, -1), (x_{p2}, -1), (x_{n2}, 1)\}$ produces confusing gradient updates.
>
>
> Q4: How does this label noise generated?
> -----------------
> **A4:** The label noise generalization method is described in the paragraph titled as "Noisy labels generation" at the bottom of page 7. We will elaborate on it in Figure 2.
>
> Q5:  "Apparently, at the instance level, the one-hot prediction of an instance is a more robust metric than the loss because the former has a tolerance to the change of predicted class posterior while the latter has not". But this doesn't appear apparent to me.
> -----------------
> **A5:** If the change of one metric is smaller than another given the same change of the output of the DNN, i.e., the predicted class posterior, we claim this metric is more robust. The one-hot prediction remains unchanged if the position of the max element in the predicted class posterior vector is maintained but the cross-entropy loss changes
> once the predicted class posterior changes. Therefore, the one-hot prediction of an instance is a more robust metric than the loss.
>
> Q6: "high noisy-TCP instances naturally indicate the instance is learned better and faster for predicting y, leading to stable and fast learning.". But I cannot see the reason here. Why is this premise natural?
> ------------------
> **A6:** High noisy-TCP by definition indicates the prediction of one instance is fixed in the very early stage, which implies the instance is learned better and faster for predicting y.
>
> Q7: Why are baselines such as Dividemix missing? Why a new method is introduced (TCP-D)?
> ------------------
> **A7:** As explained in the paper: "On synthetic noisy datasets, without introducing data augmentation techniques and semi-supervised learning, we compare the proposed method TCP with the following baselines:", therefore Dividemix is excluded on synthetic noisy datasets. To pursue SoTA performance, we apply our transition matrix learning and fine-tuning parts to the SoTA method Dividemix, which is TCP-D.

---

> > ### Author Response · Authors · 2022-11-29
> > **Sincerely looking forward to your reply.**
> >
> > We appreciate your constructive comments on our paper. We hope we have the opportunity to see your further comments, answer any additional questions, and ultimately improve the quality of our submission. Are there unclear explanations? We could further clarify them.
> >
> > Best wishes,
> > Authors

---

### Decision · Program_Chairs · 2023-01-20

**Decision:**

Reject

**Justification For Why Not Higher Score:**

There is no champion for this paper and I personally don't think this is good enough work. It's confusing and unclear in places.

**Justification For Why Not Lower Score:**

N/A

**Metareview: Summary, Strengths And Weaknesses:**

The paper discusses the issue of noisy labels in the train data, in particular when this noise can be dependent on the input. The paper motivates its approach based on observations about the difficulty of estimating a noise transition matrix, and creates a technique to monitor consistency of predictions during gradient updates. The experimental results are quite strong, though there might have I think been more work done on toy problems to demonstrate the approach better. I think the reviewers had some issues with the clarity of the paper, which I agree with and post rebuttal the review scores generally remained below threshold. There are also some statements that I find difficult to accept -- for example, in contrast to what the authors state, learning the noise transition doesn't require explicit knowledge of the clean label. There are works that treat the clean label as an unknown latent variable and, based on an assumption that noise will generally have a higher chance (for binary labels) of staying in the correct class, this allows the noise process to be identified. I think some of the basic explanations and assumptions in this work are not fully clear or justified. In general, it is also quite strange (to me at least) why a temporal criterion is used to fix an issue (label noise) that has itself nothing to do with learning dynamics.